# Biodiversity of Algae and Cyanobacteria in Biological Soil Crusts Collected Along a Climatic Gradient in Chile Using an Integrative Approach

**DOI:** 10.3390/microorganisms8071047

**Published:** 2020-07-14

**Authors:** Elena Samolov, Karen Baumann, Burkhard Büdel, Patrick Jung, Peter Leinweber, Tatiana Mikhailyuk, Ulf Karsten, Karin Glaser

**Affiliations:** 1Applied Ecology and Phycology, Institute of Biological Sciences, University of Rostock, Albert-Einstein-Straße 3, 18059 Rostock, Germany; ulf.karsten@uni-rostock.de; 2Faculty of Agricultural and Environmental Science, Soil Science, University of Rostock, Justus-von-Liebig-Weg 6, 18051 Rostock, Germany; karen.baumann@uni-rostock.de (K.B.); peter.leinweber@uni-rostock.de (P.L.); 3Plant Ecology and Systematics, University of Kaiserslautern, Erwin-Schrödinger-Straße 13, 67663 Kaiserslautern, Germany; buedel@bio.uni-kl.de; 4Applied Logistics and Polymer Sciences, University of Applied Sciences Kaiserslautern, Carl-Schurz Straße 10-16, 66953 Pirmasens, Germany; patrick_jung90@web.de; 5M.G. Kholodny Institute of Botany, National Academy of Sciences of Ukraine, Tereschenkivska Str. 2, Kyiv 01004, Ukraine; t-mikhailyuk@ukr.net

**Keywords:** biocrust, Chile, eukaryotic algae, cyanobacteria, integrative approach, climate gradient

## Abstract

Biocrusts are associations of various prokaryotic and eukaryotic microorganisms in the top millimeters of soil, which can be found in every climate zone on Earth. They stabilize soils and introduce carbon and nitrogen into this compartment. The worldwide occurrence of biocrusts was proven by numerous studies in Europe, Africa, Asia and North America, leaving South America understudied. Using an integrative approach, which combines morphological and molecular characters (small subunit rRNA and ITS region), we examined the diversity of key biocrust photosynthetic organisms at four sites along the latitudinal climate gradient in Chile. The most northern study site was located in the Atacama Desert (arid climate), followed by open shrubland (semiarid climate), a dry forest region (Mediterranean climate) and a mixed broad leaved-coniferous forest (temperate climate) in the south. The lowest species richness was recorded in the desert (18 species), whereas the highest species richness was observed in the Mediterranean zone (40 species). Desert biocrusts were composed exclusively of single-celled Chlorophyta algae, followed by cyanobacteria. Chlorophyta, Streptophyta and cyanobacteria dominated semiarid biocrusts, whereas Mediterranean and temperate Chilean biocrusts were composed mostly of Chlorophyta, Streptophyta and Ochrophyta. Our investigation of Chilean biocrust suggests high biodiversity of South American biocrust phototrophs.

## 1. Introduction

Biological soil crusts (biocrusts) are top-soil communities composed of many groups of organisms: bacteria, cyanobacteria, microalgae, microfungi, mosses, lichens, liverworts, protists and invertebrates [1]. Closely connected to soil particles these communities provide vital ecological functions in soil ecosystems—they promote nutrient cycling, increase soil stability, reduce evaporation and increase soil moisture [2]. On a global scale, biocrusts contribute between 40% and 85% to biological N fixation by terrestrial organisms and with 15% to the global net terrestrial primary production [3]. Biocrusts are often pioneer communities in ecosystems degraded by natural or manmade activities (e.g., habitats affected by fire events or glacier retreat, mining areas, etc.), where they induce soil formation and facilitate the restoration of native vegetation [4,5].

One of the crucial factors for the ecosystem’s stability and productivity is the variety of taxa in a natural community. Since phototrophic microorganisms have different, species-specific ecophysiological requirements, i.e., different tolerances against environmental factors, with higher biodiversity the community will have broader tolerance to various abiotic conditions. In addition, a higher number of species can simultaneously provide multiple functions in a biocrust, which will result in the more productive community [6,7]. Hu et al. [8] found that artificial single-species biocrusts were not as effective in soil aggregation as the multispecies crusts. Higher diversity of organisms promotes the resilience of biocrusts to environmental changes without losing the potential to recover to an initial state, which in turn increases the ecosystem’s resilience and recovery potential [9].

Biogeographical distribution of species depends on several factors: population size, the efficiency of distribution mechanisms and the availability of suitable habitats. Microalgae and cyanobacteria can be transported to varying distances by air or by moving animals (birds, insects, etc.), yet an efficient dispersal in aero-terrestrial conditions also depends on the resistance of species to survive unfavorable conditions during transport (e.g., desiccation and intensive UV radiation [10]). Aero-terrestrial cyanobacteria and algae have numerous water-holding mechanisms, such as the production of intracellularly accumulated desiccation-protective metabolites (e.g., polyols) or extracellular, thick gelatinous layers and mucilaginous extracellular polymeric substances (EPS), which guarantee more efficient dispersal compared to their aquatic counterparts [10,11]. The availability of favorable habitat conditions is another crucial factor for the occurrence and future role of a taxon in a community. The life-history causes specialists to be niche- and habitat-limited, while generalists are mainly limited by the efficiency of dispersion [12]. Once established in the favorable habitat well-adapted taxa will be the most abundant and mostly predated in the biocrust community, thus these key taxa become the major contributors to nutrient and energy flows in this (micro) ecosystem. On the other hand, rare taxa, which are most of the time in a dormant state or growing extremely slowly, will be present in a lower number and will be exposed to reduced predation pressure [13].

Biocrusts are present in every climate zone on Earth [14]. They are the dominant vegetation type in regions where higher plants are limited by low water contents and/or nutrient supply, such as semiarid and arid areas. Distribution of biocrusts on a global scale depends mainly on water availability and the topography of a habitat. While biocrusts represent typical vegetation of dryland areas, which occupy 41.3% of the Earth’s surface [15], these communities can also be found in other regions such as temperate sand dunes and disturbed habitats [14,16,17,18].

Even though biocrusts are present in every climate zone, these communities are unequally studied on a global level. Most information on biocrust organisms comes from Europe, North America, Asia and Australia. However, biodiversity hotspots, such as Africa and South America, are only little, or not investigated at all. Büdel et al. [19] showed a strong discrepancy between continental size and research focused on biocrust algae. Studies focused on selected geographical areas made Europe and Antarctica have higher species’ richness than South America. Biocrust-forming prokaryotic and eukaryotic algae are almost unstudied in South America. According to Büdel et al. [19], only one algal and 40 cyanobacterial species were reported from South American biocrust communities, compared to 43 algal and 65 cyanobacterial species reported from polar biocrusts of Antarctica.

First records of “cryptogamic vegetation” in Chile were studies by Schwabe [20] and Forest & Weston [21], which focused exclusively on the diversity of cyanobacteria and were limited to the arid locations in northern Chile. The first report of eukaryotic algae in Chilean soil was a study by Patzelt et al. [22]. The authors investigated the hyperarid core of the Atacama Desert and the semiarid areas in the vicinity of the Nature Reserve Santa Gracia. This study presented detailed molecular identification of cyanobacteria but did not provide any information on the eukaryotic algae that were found. Several new cyanobacteria were recently described from the Atacama Desert: Kastovskya, Symplocastrum, Myxacorys, Trichocoleus and Oculatella species [23,24,25,26,27]. The most recent is a study by Jung et al. [28] which investigated the diversity of hypolithic and edaphic cyanobacteria in two study areas, hyperarid Pan de Azúcar and the semiarid area of the Nature Reserve Santa Gracia; the authors reported 21 cyanobacterial species and the new species *Aliterella chasmolithica* [29].

To which extent the biocrust flora of South America is understudied was shown by the comprehensive publication of Büdel et al. [19] who reviewed 39 scientific papers on a diversity of biocrust algae found globally. Out of reported 360 species of eukaryotic algae, the only taxon found in South America was *Zygogonium ericetorum*. A new species and new varieties of rare green alga Pleurastrosarcina were described recently from a biocrust of National Park Pan de Azúcar and Nature Reserve Santa Gracia [30]. The first insight into the diversity of biocrust cyanobacteria and algae along the climatic gradient in Chile was provided by Baumann et al. [4]. This study, however, was based on the identification of organisms on morphological traits only. In a more recent study by Samolov et al. [31] four new species and two varieties of the filamentous genus Klebsormidium (Streptophyta) were described based on 19 strains isolated from the Chilean biocrusts. Klebsormidium is one of the key taxa in biocrusts worldwide [19,32].

Since all previous studies of terrestrial cyanobacteria and eukaryotic algae were focused exclusively on the arid area of the Atacama Desert, the objective of present study was to document in detail the diversity of key biocrust photosynthetic microorganisms in study areas which follow the precipitation gradient along the Pacific coast of Chile. We hypothesized that the precipitation gradient shaped the biocrust photosynthetic microorganism communities, i.e., the drier the habitat the lower the species richness. For the first time, Chilean biocrust cyanobacteria and eukaryotic algae were analyzed and identified based on both morphological and molecular characters.

## 2. Materials and Methods

### 2.1. Study Site

Biocrust samples were collected from all four study areas of the EarthShape priority program funded by the German Research Council (DFG) [4]. The sampling locations were situated within a north–south precipitation gradient, with National Park Pan de Azúcar (PA) in the arid north at 651 m above sea level (ASL), Nature Reserve Santa Gracia (SG) in the semiarid zone at 706 m ASL, National Park La Campana (LC) in the Mediterranean climate zone at 726 m ASL and National Park Nahuelbuta (NB) at 770 m ASL in the temperate-humid zone of Chile (Figure 1).

Prominent latitudinal precipitation gradient along the Pacific coast of Chile caused distinct climate and vegetation zones. From the cold coastal Atacama Desert in the North with 14.7 mm·y^−1^ in Pan de Azúcar (PA) mean annual precipitation gradually increases towards the south, reaching 934.3 mm·y^−1^ in Nahuelbuta (NB). Located in the semiarid area Santa Gracia (SG) receives 83.4 mm·y^−1^, while Mediterranean La Campana (LC) receives 314.5 mm·y^−1^ [4]. The aridity in PA and SG is alleviated by the coastal fog which stretches inland via fluvial channels [33] and increases water deposition by 1.4 L m^−2^ d^−1^ in PA and 3.0 L m^−2^ d^−1^ in SG [34].

Higher vegetation reflects the precipitation gradient along the Chilean coast. Vegetation in desert PA is scarce, limited by a strong water deficit. The landscape is dominated by biocrusts, which cover up to 40% of soil and by sporadic small bush vegetation. In semiarid SG, vegetation is composed of drought-deciduous shrubs, cacti and biocrusts. Biocrusts, which cover up to 15% of the soil surface, were under strong mechanical disturbance by livestock grazing and trampling. As the mean annual precipitation increases, it promotes the development of higher plants, which in turn has a negative effect on the areal coverage of biocrusts. In both Mediterranean LC and temperate, forested NB, biocrust covered only up to 5% of soil surface. In these areas, biocrusts are limited to locations where the dominant plant cover is disrupted by natural or manmade disturbances (data presented in Bernhard et al. [35]).

Desert biocrusts of PA were composed of predominantly chlorolichens and green algae, with cyanobacteria being present sporadically, and bryophyte being absent from the community. Biocrust communities in semiarid SG were showing a prevalence of chlorolichens compared to bryophyta. Cyanobacteria were the dominant component in these communities when compared to eukaryotic algae. Compared to other biocrusts along the precipitation gradient, Mediterranean LC communities were predominantly composed of eukaryotic algae, followed by cyanobacteria and bryophyta; chlorolichens represented a very low portion of the community. In the temperate region of Nahuelbuta higher plants dominate the landscape, biocrusts are dominated by bryophyta and eukaryotic algae, while the species’ richness of *N*-fixing cyanobacteria decreases. In these communities no chlorolichens were present (data are shown in Bernhard et al. [35]).

Biocrust material was collected during sampling campaigns in January and March 2016. Samples were collected according to the sampling procedure described in detail in Schulz et al. [36]. In short, two biocrust samples from each study site were taken from the upper, topsoil layer (<5 mm) by pushing a spatula gently below the biocrust. The spatula with the biocrust was then lifted, and the sample was carefully transferred into a small paper box. These samples had a surface area of approximately 6 × 6 cm and a thickness of <5 mm. The collected material was air-dried for the transport, then kept in the laboratory at room temperature in the dark. Biocrust material was used for the establishment of enrichment cultures within three weeks after the sampling.

### 2.2. Cultivation of Strains and Morphological Identification

In order to identify biocrust-forming photosynthetic organisms, an integrative approach was applied. Small amounts of biocrust material (approximately 1 × 1 cm) were inoculated on solid 3N Bold’s Basal Medium (3 N BBM) with 1.5% Difco agar (Fisher Scientific, Schwerte, Germany) [37] to promote the growth of eukaryotic algae. The same inoculation method was applied for the establishment of enrichment cultures on the solid BG11 agar plates, to promote the development of cyanobacteria from the biocrust community. Diatoms were not investigated in the present study since the applied methods of cultivation and identification (light microscopy) were not appropriate for this algal group (e.g., Schulz et al. [36]).

Enrichment cultures were incubated at room temperature, under a photon fluence rate of approximately 30-µmol photons m^−2^ s^−1^ (light source: Osram Daylight Lumilux Cool White lamps (L36 W/840); Osram, Munich, Germany) and a 16:8 h light–dark cycle [36]. After 4-week cultivation single colonies of algae or cyanobacteria could be observed, which were then transferred to new 3-N BBM or BG11 [38] agar plates to establish pure clonal cultures; the unialgal or unicyanobacterial cultures were cultivated under the same conditions.

Morphological characters of every isolated biocrust alga or cyanobacterium were examined with a stereoscopic microscope Olympus SZX16 and Olympus BX51 light microscope with Nomarski DIC optics (Olympus Optical Co., Ltd., Tokyo, Japan) at 1000-fold magnification (oil immersion). Photomicrographs were taken with digital camera Olympus UC30 (Olympus Soft Imaging Solutions, Münster, Germany) attached to the microscope and processed with the Olympus cellSens Entry software (v. 2.1, Olympus Soft Imaging Solutions, Münster, Germany). Morphological identification of eukaryotic algae was based on Ettl & Gärtner [39] and the latest scientific literature on specific taxa; morphological identification of cyanobacteria was based on Komárek & Anagnostidis [40] and Komárek [41] as well as latest scientific literature on specific cyanobacterial taxa.

### 2.3. DNA Extraction, PCR

DNA of selected algal strains was extracted using the DNeasy Plant Mini Kit (Qiagen, Hilden, Germany) according to the manufacturer’s instructions. Nucleotide sequences of small subunit (SSU) rRNA gene together with ITS-1-5.8S-ITS-2 region of Chlorophyta and Streptophyta were amplified using a set of Taq PCR Mastermix Kit (Qiagen, Hilden, Germany), a complex of EAF3, ITS055R [42,43] and algal-specific primers G500 F and G800R [30]. The PCR was conducted according to Mikhailyuk et al. [44]. The PCR products were cleaned with the SureClean Plus purification kit (Bioline, Luckenwalde, Germany) according to the manufacturer’s instructions. Cleaned PCR products were sequenced commercially by Qiagen Company (Hilden, Germany) using primers EAF3, 536R, 920F, 1400R, GR and GF [43,45,46,47]. The resulting sequences were assembled and edited using Geneious software (v. 8.1.8; Biomatters). They were deposited at GenBank under the accession numbers MT735190-MT735212 and MT738691-MT738694.

Total genomic DNA of cyanobacterial isolates was extracted by the cetrimonium bromide (CTAB) method followed by phenol-chloroform-isoamyl alcohol purification adapted for biocrusts [48]. A nested PCR approach was chosen for a first PCR with the primer set 27F1 and 1494Rc followed by a subsequent second PCR with the primer set CYA361f and CYA785r for cyanobacteria [49], with an annealing temperature of 61 °C. The obtained PCR products were cleaned by using the NucleSpin^®^ Gel and PCR Clean-up Kit(Macherey-Nagel GmbH & Co. KG, Düren, Germany). Cleaned PCR products were sequenced by Seq-It GmbH & Co. KG (Kaiserslautern, Germany), and the sequences were submitted to GenBank (Accession No. MT702999-MT703004).

### 2.4. Phylogenetic Analyses

Sequences of the algal and cyanobacterial isolates were compared to those from reference strains in NCBI (http://www.ncbi.nlm.nih.gov) using BLASTn queries [50] to find the closest relatives; the NCBI database was searched for algae on 22 January and for cyanobacteria on 17 February 2020. When possible we used sequences of authentic algal and cyanobacterial strains, on which the formal taxonomic description of a specific species was based. We included also edaphic cyanobacteria from SG and PA described by Jung et al. [51] for our analysis. Multiple alignments of nucleotide sequences of the SSU rRNA were made with the Mafft web server (v. 7 [49]), followed by manual editing in the program BioEdit (v. 7.2). Alignments for the phylogeny of the ITS-1,2 were performed manually in BioEdit, taking into account the secondary structure of the RNA in the region. The evolutionary model that was best suited to the database used was selected based on the lowest AIC value [52] and calculated in MEGA (version 6 [53]). Phylogenetic trees were constructed in the program MrBayes 3.2.2 [54], using an evolutionary model GTR + G + I, with 5,000,000 generations. Two of the four runs of the Markov chain Monte Carlo were made simultaneously, with the trees, taken every 500 generations. Split frequencies between runs at the end of the calculations were below 0.01. The trees selected before the likelihood rate reached saturation were subsequently rejected. The reliability of tree topology was verified by maximum-likelihood analysis (ML), using the program GARLI 2.0 [55], and the bootstrap support was calculated with 1000 replicates.

## 3. Results

### 3.1. Diversity of Algae—Identification Based on Morphology and Molecular Phylogeny

The total species list of algae includes 63 taxa belonging to infrakingdom Chlorophyta, phylum Chlorophyta (45 species among classes Chlorophyceae (23) and Trebouxiophyceae (22)), Streptophyta (13 species, classes Klebsormidiophyceae (11) and Zygnematophyceae (2)) and Ochrophyta (5 species, classes Xanthophyceae (3) and Eustigmatophyceae (2)). This list was obtained based on the culture-dependent approach in combination with morphological traits (Table 1). Some of these strains were additionally analyzed using molecular markers.

Phylogenetic analysis of SSU rRNA included 21 strains of algae from classes Chlorophyceae and Trebouxiophyceae.

Chlorophyceae strains were distributed between the two main groups: Chlamydomonadales: genera Lobochlamys (Oogamochlamydinia), Fasciculochloris and Heterochlamydomonas (Reinhardtinia), Ixipapillifera (Chloromonadinia), Tetracystis (Dunaliellinia); and Spaheropleales (genus Bracteacoccus).

Lobochlamys strains corresponded to *L. segnis* and formed a separate lineage inside Oogamochlamydinia which was impossible to identify as known genus or species. Strain 4SG-4 could represent a species of Fasciculochloris, but it was distant from the authentic strain *F. boldii*. One of the Heterochlamydomonas strains was close to *H. inaequalis*, another one formed a separate lineage inside this genus. A Tetracystis strain from Dunaliellinia was close to the authentic strains of *T. intermedia* and *T. pulchra*. Bracteacoccus strains corresponded to *B. bullatus* and formed a separate unidentified lineage inside the genus (Figure 2).

Trebouxiophyceae strains were distributed among the genera Watanabea, Coccomyxa, Elliptochloris, Parietochloris, Myrmecia, Stichococcus, Gloeocystis and Edaphochlorella (Figure 3). Watanabea, Coccomyxa, Elliptochloris and Edaphochlorella strains were close to the authentic strains of *W. borystenica*, *C. simplex, E. perforata* and *E. mirabilis.* Barcoding region of SSU sequence of Elliptochloris strain (PA-1–3) was identical to authentic strains of *E. perforata* [56], and hence it was identified as respective species. Parietochloris, Myrmecia, Stichococcus, and Gloeocystis strains showed unresolved positions inside respective genera, and indicated taxonomic revisions which were not proposed so far.

Phylogenetic analyses of ITS region were performed for the genera Bracteacoccus, Coccomyxa and Watanabea (Figure 4), for which respective taxonomic revisions based on ITS phylogeny were proposed. These deeper analyses allowed us to identify strains to species level as *B. bullatus*, *C. simplex* and *W. borystenica*. ITS region of Tetracystis strain was the most similar to *T. intermedia* with few nucleotide differences (3 nucleotide differences in ITS1 and 2 nucleotide differences in ITS2), therefore it was possible to identify the strain as the respective species. Comparison of ITS sequence of Edaphochlorella strain with the authentic strain of *E. mirabilis* showed a few nucleotide differences in ITS2, which was located outside of the barcoding region [56]. Therefore, the identification of this strain as *E. mirabilis* was confirmed.

Representatives of class Klebsormidiophyceae were analyzed based on ITS 1, 2 phylogeny together with the sequences of Chilean strains presented in Samolov et al. [31]. Twenty-three Klebsormidiophyceae strains included 4 representatives of the genus Interfilum and 19 strains of the genus Klebsormidium (Figure 5). Eight strains, which belong to clade E, were identified as *Klebsormidium* sp., *K. fluitans* and *K. nitens*. Eleven strains were representatives of clade G. They were described as new species and varieties: *K. deserticola*, *K. chilense*, *K. sylvaticum*, *K. delicatum* var. *americanum* and *K. delicatum* var. *deserticum* [31]. Interfilum strains were clustered with isolates of *I. massjukiae* as well as formed two separate lineages, which did not correspond to any known species.

The analysis of only morphological characters of Chilean biocrust algae enabled the determination of 12 taxa to the species level: *Bracteacoccus medionucleatus, Chlorococcum echinozygotum, Pseudomuriella aurantiaca,* etc. Since morphological characters of certain strains were insufficient for precise identification, 15 strains were tentatively identified: *Chlamydomonas* cf. *pseudoelegans*, *Chlorococcum* cf. *minimum*, *Chlorococcum* cf. *minutum*, *Chlorococcum* cf. *oleofaciens*, etc. Identification of some of these taxa was confirmed by molecular phylogenetic method.

Strains, which lacked distinctive morphological characters for the identification to the species level, were identified to genus level only: *Chlamydomonas* sp.1, *Chlamydomonas* sp.2, *Chlorococcum* sp.1, *Chlorococcum* sp.2, *Fasciculochloris* sp., Taxonomic revisions based on molecular phylogeny for these taxa are still missing. Morphological characters of isolated algae from investigated localities are presented in Figure 6, Figure 7, Figure 8 and Figure 9.

Preliminary list of green algae from investigated biocrusts identified using morphological methods was published in Baumann et al. [4]. Using molecular methods as well as more detailed morphological investigation allowed us to redefine some taxa: *Gloeocystis* sp. (preliminary determined as *Coenochloris signiensis*), *Elliptochloris perforata* (*E. bilobata*), *Bracteacoccus* sp. (*Nannochloris* sp.), *Neocystis* cf. *brevis* (*Neocystis* sp.), *Parietochloris pseudoalveolaris* (*P. alveolaris*), *Tetracystis intermedia* (*T. compacta*), several new taxa of clade G *Klebsormidium* described and published in Samolov et al. [31] (*Klebsormidium* sp.).

### 3.2. Diversity of Cyanobacteria—Identification Based on Morphology and Molecular Phylogeny

Species list of cyanobacteria includes 24 species which belonged to orders Chroococcidiopsidales (1 species), Nostocales (5), Oscillatoriales (8), Pleurocapsales (1) and Synechococcales (9). This list was obtained by the culture-dependent method and integrative approach for identification of cyanobacteria (Table 2).

Some strains were additionally analyzed with molecular markers. Phylogenetic analysis of SSU region included 16 strains of cyanobacteria from the orders Nostocales, Oscillatoriales, Pleurocapsales and Synechococcales (Figure 10).

Some strains were already mentioned in Jung et al. [51]. Nostocales strains were clustered with species of Nostoc, Oscillatoriales strains fell in the clade formed by isolates identified as *Trichocoleus sociatus* and *Microcoleus vaginatus*. Pleurocapsales strain was clustered close to the isolate of *Pleurocapsa minor*. Synechococcales strains fell in clades of Nodosilinea and Stenomitos. Nodosilinea and Trichocoleus strains were close to authentic strains of *N. epilithica* and *T. badius*, therefore they were identified as respective species. Two strains identified by us as *T. sociatus* were clustered together with a group of Oscillatorialean strains (“*Trichocoleus sociatus*” group). The group potentially corresponds to Microcoleus sensu lato because it was situated outside of the Synechococcales to which Trichocoleus belongs (see Mühlsteinová et al. [24]). Strains of Nostoc and Stenomitos had unresolved positions since taxonomic revisions of these genera is still in progress, therefore, it is impossible to assign these strains to known species.

Based on morphological characters we identified 4 cyanobacterial strains to the species level as *Leptolyngbya henningsii*, *Leptolyngbya tenuis*, *Microcoleus vaginatus* and *Nodosilinea epilithica*. In cases where the identification to species level was not possible based on morphology we identified 5 strains tentatively as *Nostoc* cf. *edaphicum*, *Nostoc* cf. *punctiforme*, *Oscillatoria* cf. *tenuis,* etc. Thirteen strains were identified to the genus level: *Chroococcidiopsis* sp., *Nostoc* sp., *Microcoleus* sp., *Phormidium* sp., *Myxacorys* sp., etc.

Preliminary list of cyanobacteria from investigated biocrusts identified using morphological methods was published in Baumann et al. [4]. Using molecular methods as well as more detailed morphological investigation allowed to redefine some taxa: *Nostoc* sp. 1 and sp. 2 (preliminary determined as *Nostoc commune*) and *Trichocoleus badius* (*T. desertorum*).

### 3.3. Biocrust Community Composition and Species Richness

With regard to the culture-based community composition and species richness assessment, desert biocrusts from Pan de Azúcar (PA) were composed exclusively of Chlorophyta representatives, with a clear dominance of Trebouxiophyceae (9 species) over Chlorophyceae (1 species). In semiarid Santa Gracia (SG) and Mediterranean La Campana (LC) communities were dominated by representatives of Chlorophyta (9 species in SG and 24 species in LC) and Streptophyta (2 species in SG and 5 species in LC), followed by Ochrophyta (3 species in SG and 4 species in LC) (Figure 11).

In both SG and LC biocrusts Chlorophyceae (5 species in SG versus 14 species in LC) outnumbered Trebouxiophyceae (4 species in SG versus 10 species in LC). While LC biocrusts had a higher diversity of Xanthophyceae (3 species), SG had a slightly higher diversity of Eustigmatophyceae (2 species). Nahuelbuta (NB) biocrusts were composed of Chlorophyta, Streptophyta and Ochrophyta representatives. In this biocrust community, 17 Chlorophyta species were members of classes Trebouxiophyceae (10 species) and Chlorophyceae (7 species). Temperate NB biocrusts had the highest diversity of Streptophyta representatives (6 species) when compared to other biocrusts we analyzed.

Cyanobacterial representatives in PA biocrusts were members of Chroococcidiopsidales (1 species), Nostocales (1), Oscillatoriales (2), Pleurocapsales (1) and Synechococcales (2). SG biocrusts were composed of Nostocales, Oscillatoriales and Synechococcales. The highest diversity was observed in Oscillatoriales (6 species), followed by Synechococcales (4) and Nostocales (3). LC communities had a high diversity of Synechococcales (4 species), followed by Nostocales (2) and 1 species from Oscillatoriales. Biocrusts from NB were formed by Oscillatoriales (2 species) and one Nostocales representative.

The total number of algal species was lowest in Pan de Azúcar biocrusts (11 species) and highest in the biocrusts from La Campana (33 species). A higher number of algal species was found in the Nahuelbuta biocrusts compared to Santa Gracia (24 and 14 species).

The total number of cyanobacterial species was lowest in NB (3), followed by LC and PA biocrusts (7). The highest number of cyanobacterial species was observed in SG (12) biocrusts.

Single-celled or packet-like (coccoid) eukaryotic algae (Table 1) dominated all Chilean biocrusts. The only filamentous algae were representatives of genera Klebsormidium (Streptophyta) and Xanthonema (Ochrophyta). Klebsormidium representatives were found in all biocrust communities, except in the desert ones (PA); *Xanthonema exile* was present in LC biocrusts only.

Filamentous cyanobacterial species dominated biocrust communities from LC, SG and NB, representatives of genera Microcoleus, Myxacorys, Leptolyngbya and Stenomitos. Only arid PA communities were dominated by thallus-forming, colonial cyanobacteria such as representatives of genus Nostoc and Chroococcidiopsis (Table 2). A species, which was present in all investigated biocrusts, was *Diplosphaera chodatii*.

## 4. Discussion

In the present study, we are showing for the first time a comprehensive analysis on the biodiversity of biocrust-forming microalgae and cyanobacteria in four climate zones in Chile, identified by an integrative approach, which takes into account both morphological and molecular traits. It should be taken into account that in our study community composition and species richness assessment is based on the culture-dependent approach, which can underestimate algal and cyanobacterial diversity in natural communities. The culture-depended approach can potentially lead to the overestimation of algal and cyanobacterial species, which can grow fast in culture. It can also lead to underestimation or even failing to detect uncultivable species, which could be dominant elements in natural biocrusts [19,36].

Biocrust communities we analyzed were predominately composed of algae from the classes Chlorophyceae and Trebouxiophyceae, followed by Klebsormidiophyceae, which corresponds to previous studies on biocrusts from Europe, Asia, Africa, North America, Australia and Polar Regions of Russia presented by Büdel et al. [19]. Algal species richness in Chilean biocrusts (45), when compared to Australia (3) or Antarctica (44) as presented in Büdel et al. [19], reflects the general lack of information on the diversity of biocrust algal globally.

Depending on their morphology and functional traits, biocrust algae can be divided into biocrust-forming, filamentous and biocrust-associated, coccal taxa. Chilean biocrust algae are phylogenetically diverse, the majority were unicellular or strains which form colonies with densely packed cells. The coccal biocrust algae can be attached to filamentous forms, soil particles or they can live epiphytically on the thallus of biocrust lichens or mosses [19,36]. Chilean unicellular algae were present in lower biomass in biocrusts, but their diversity exceeded the diversity of filamentous representatives. This observation was consistent with the previous analysis of biocrust communities worldwide [19].

We found several algal species, which were new to South America and even represented completely new species. The data set shown thus proves our presumption that algal species described from South America are strongly underestimated compared to Europe or North America due to less sampling activities. For example, all filamentous algae in Chilean biocrusts belonged to the genus Klebsormidium and they were present in all biocrust samples, except the desert PA communities. Most isolated Klebsormidium strains were representatives of clade G, which was originally designated as desert clade [57]. With the newly isolated Klebsormidium clade G strains, it was possible to describe four new species, including representatives from the Mediterranean and temperate areas of Chile [31]. The discovery of South American strains from Klebsormidum clade G broadened the ecological distribution of this clade to semiarid, Mediterranean and temperate climate zone. This example shows that studies in South America have the potential to discover new microalgal species even in a well-known genus such as Klebsormidium.

Another interesting finding was a microalga (strain LC006-5), which could represent a new lineage inside Oogamochlamydinia (confirmed by molecular methods). This isolate should be investigated taxonomically much deeper to clarify if it represents a new species or even a new genus. This example emphasizes the necessity for additional thorough taxonomic studies to reliably describe the biodiversity of microalgae in South America.

Several species in the Chilean biocrust belong to just recently described or revisited taxa. For example, Ixipapillifera is a newly erected genus of flagellated algae with specific X-shaped papilla. Members of this genus are typically found in soil (*I. deasonii*) or freshwater habitats [58].

Another interesting finding is the widely distributed *Xerochlorella minuta* recently revised in Mikhailyuk et al. (2020) [59]. This alga, usually known as *Dictyosphaerium minutum*, is a typical component of biocrusts; it prefers xerophytic habitats, which vary from Polar Regions to maritime sand dunes and hot deserts.

Two strains isolated from biocrusts of LC were identified as *Watanabea borystenica*, a rare and interesting genus [60]. The authentic strain of this terrestrial algal species was just recently described and isolated from acidic soil deposited after coal mining in Sokolov mining district, Czech Republic [60]. According to the type locality, *W. borysthenica* can inhabit open habitats with intensive solar radiation and unfavorable water regime. Chilean strains are the first record of Watanabea genus being part of a Mediterranean biocrust community. This finding indicates that in South America it is possible to find species, which are rare in Europe and were found so far in ecologically disturbed habitats like extremely acidic soils.

### 4.1. Differences in Community Composition of Biocrust Algae and Cyanobacteria along the Precipitation Gradient

We expected the effect of the precipitation gradient to be reflected in the algal and cyanobacterial species composition of Chilean biocrusts. Indeed, we observed clear differences between the four regions. In the arid PA, biocrusts dominate the landscape since they can cover up to 40% of soil surface [35]. The other sampling regions are dominated by higher vegetation, limiting the area covered by biocrusts to up to 15% in SG and 5% in LC and NB.

Comparing the biocrust communities from the four regions, from the arid north to mesic south, the following pattern was observed: all algae in PA were either single-celled or colony-forming, whereas non-filamentous cyanobacteria were only detected in PA. Since PA is a desert area with nearly no rain, where fog or dew act as the only regular water sources, biocrust communities of this very specific region were separately discussed. However, scarce water input of this site enabled the development of early to climax biocrusts dominated by chlorolichens and algae, over cyanobacteria.

In the other three regions along the precipitation gradient, rain events occur, thus biocrusts could develop well. Therefore, filamentous algae and filamentous cyanobacteria were present in biocrusts in all other regions.

In semiarid SG, filamentous cyanobacteria were prevailing over algae, contrary to Mediterranean LC biocrusts, which were predominantly composed of eukaryotic algae, followed by filamentous and colonial cyanobacteria. The dominance of algae over cyanobacteria in the temperate forest NB biocrusts was observed in European communities [18,61] as well.

While filamentous cyanobacteria dominated biocrusts in the semiarid region, eukaryotic filamentous algae had the highest richness in the temperate region. Low amount of cyanobacteria in NB biocrusts is common for forest ecosystems and can be attributed to unfavorable soil pH and limited light conditions [18,62]. The dominance of better adapted filamentous algae from the genus Klebsormidium over filamentous cyanobacteria in NB and partly also in LC communities is in accordance with reports of Glaser et al. [18], where Klebsormidium was reported to “replace” cyanobacteria as a biocrust-forming alga.

### 4.2. Pan de Azúcar Biocrusts as an Example for an Extreme Biocrust Habitat

PA biocrusts corresponded to the communities observed in the Namibian fog desert. Just like in Africa, the Chilean biocrusts were devoid of cyanolichens [35], with cyanobacterial members confined to hypolithic habitats [51,63]. Jung et al. [28] described “grit crust”, a transitional form between rock cover and soil crust in a classical sense. Grit crusts are communities consisting of fungi, cyanobacteria and algae, formed on top of singular grit-sized quartz and granitoid stones, which cover and connect the grit-sized rocks. Hypolithic habitats increase the possibility of fog and dew-water condensation and shield the organisms from damaging solar radiation [64]. Lange et al. [65] concluded that cyanobacterial presence in a habitat is restricted by the liquid water availability, whereas algae can use high air humidity (i.e., fog) to be metabolically active.

Single-celled species formed loose aggregates (*Myrmecia* sp.) or the cells were protected by thick gelatinous structures (*Gloeocystis* sp.) from water loss. Like all algae from the former Radiococcaceae group, *Gloeocystis* sp. is characterized by well-developed mucilaginous layers, which protect single cells from desiccation. These layers can absorb water from scarce rain events or even high air humidity caused by the coastal fog in PA. Colony- or packet-forming algae, like *Interfilum massjukiae*, create layered structures where upper layers, usually exposed to harsh environmental conditions, protect underlying cells [66]. *I. massjukiae* was originally described from Crimea Mountains as an epilith [67] and later found as a phycobiont of lichens [68]. Colony- or packet-like morphotype is considered as an adaptation of algal cells to retain cellular water under dry conditions [69], which corresponds to the desert habitat of PA.

Cyanobacteria found in PA biocrusts include coccoid *Chroococcidiopsis* sp., which was reported as a constituent of biocrust communities worldwide [55] as well as a member of hypolithic communities in Chile [70,71,72] Jung et al. [51] reported representatives of genus Chroococcidiopsis in both edaphic and hypolithic PA communities. Cyanobacteria from this genus synthesize extracellular polymeric substances, which provide an effective defense mechanism against desiccation [19]. In addition, Chroococcidiopsis representatives are known to accumulate scytonemin, a UV-sun-screening compound extracellularly [73]. Although not studied in detail, other biocrust algae and cyanobacteria are expected to synthesize and accumulate high concentrations of UV-sunscreens to cope with the extremely high radiation conditions of the Atacama Desert [74]. *Pleurocapsa minor* is a coccoid, packet-like cyanobacteria, which was found in both edaphic and hypolithic PA communities [51]. Representatives of genus Pleurocapsa, were present in biocrusts of Namib [75] and dominating the biocrust communities in semiarid areas of Spain [76]—both with low precipitation similar to PA.

Filamentous cyanobacterial representatives *Microcoleus vaginatus*, *Pseudophormidium* cf. *hollerbachianum*, *Phormidesmis* sp. and *Trichocoleus badius* contribute to the physical stability of PA biocrusts. *Trichocoleus badius* is an interesting and poorly studied cyanobacterium which genus is typical for desert habitats [24]. The filaments are acting as supporting structures in the desert soil, which reduce the mobility of loose soil particles and promote colonization of non-motile, single-celled or colonial algae and cyanobacteria, such as Nostoc. Heterocystous Nostoc contributes to nitrogen fixation and priming of soil [76].

## 5. Conclusions

The present study is one of the few studies in South America, which provides comprehensive information on the biodiversity of microalgae and cyanobacteria isolated from biocrusts along a longitudinal climate gradient in Chile. It should be taken into account that our study is based on the culture-dependent approach, which can underestimate the algal and cyanobacterial diversity present in natural communities. The data indicate that some rather cosmopolitan biocrust key taxa such as *Microcoleus vaginatus* occur in South American biocrusts as well. On the other hand, newly identified taxa like new Klebsormidium-species suggest potentially much higher biodiversity in the Chilean biocrusts than documented in this study. Taxonomic revision for some of the new isolates is needed for an accurate biodiversity estimation. This study pointed out that biocrusts in South America represent micro-ecosystems with the potential to discover new and ecologically interesting microalgal species.

## Figures and Tables

**Figure 1 microorganisms-08-01047-f001:**
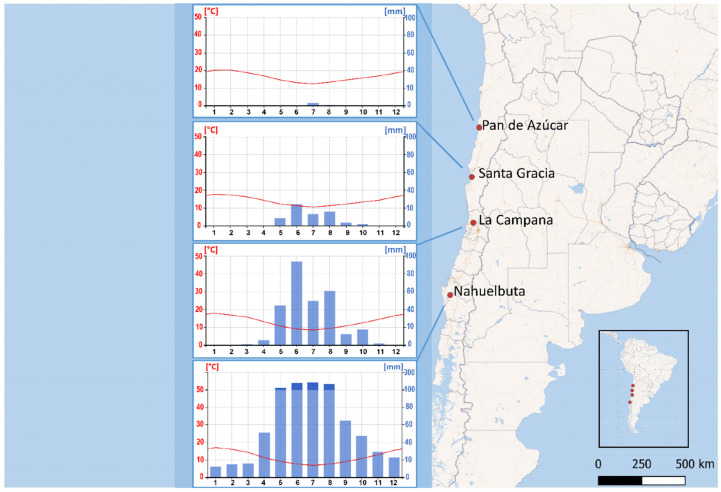
Map of the study locations in Chile with climate graph for each location. Shown are the monthly average values of diurnal temperature (°C) represented with red line, and precipitation (mm) represented with blue bars; months are represented numerically. Source: climatecharts.net, EarthShape weather data collection.

**Figure 2 microorganisms-08-01047-f002:**
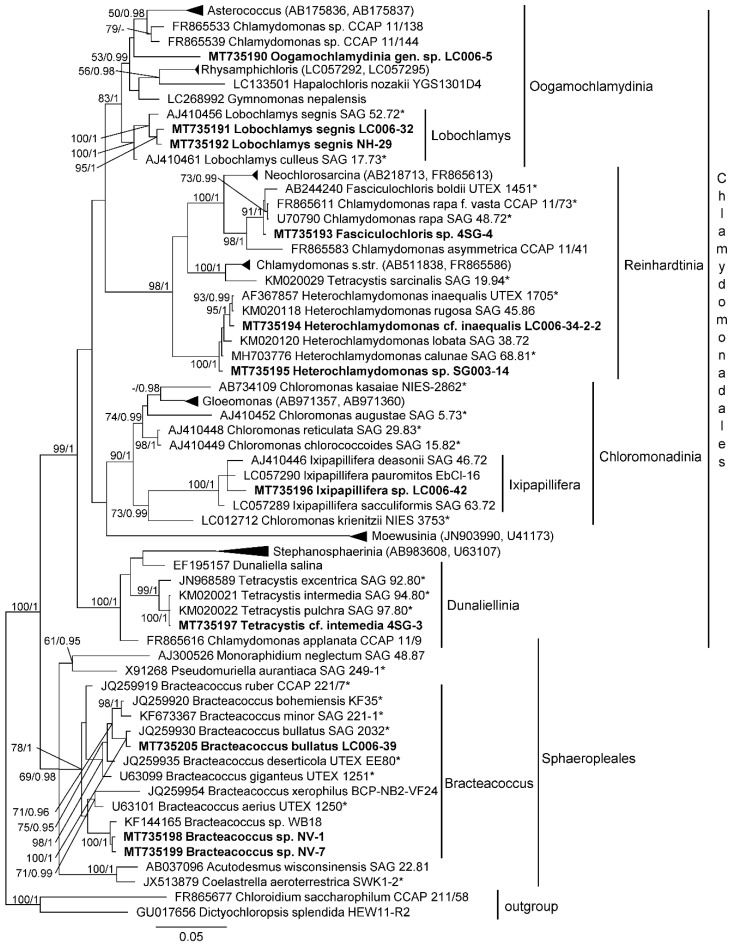
Molecular phylogeny of Chlorophyceae based on SSU rRNA sequence analysis. A phylogenetic tree was inferred by the Bayesian method with Bayesian posterior probabilities (PP) and maximum likelihood bootstrap support (BP); branches supported in both analyses (Bayesian values > 0.9 and bootstrap values > 60%) are labeled. Strains marked with an asterisk are authentic strains; strains in bold represent newly sequenced algae.

**Figure 3 microorganisms-08-01047-f003:**
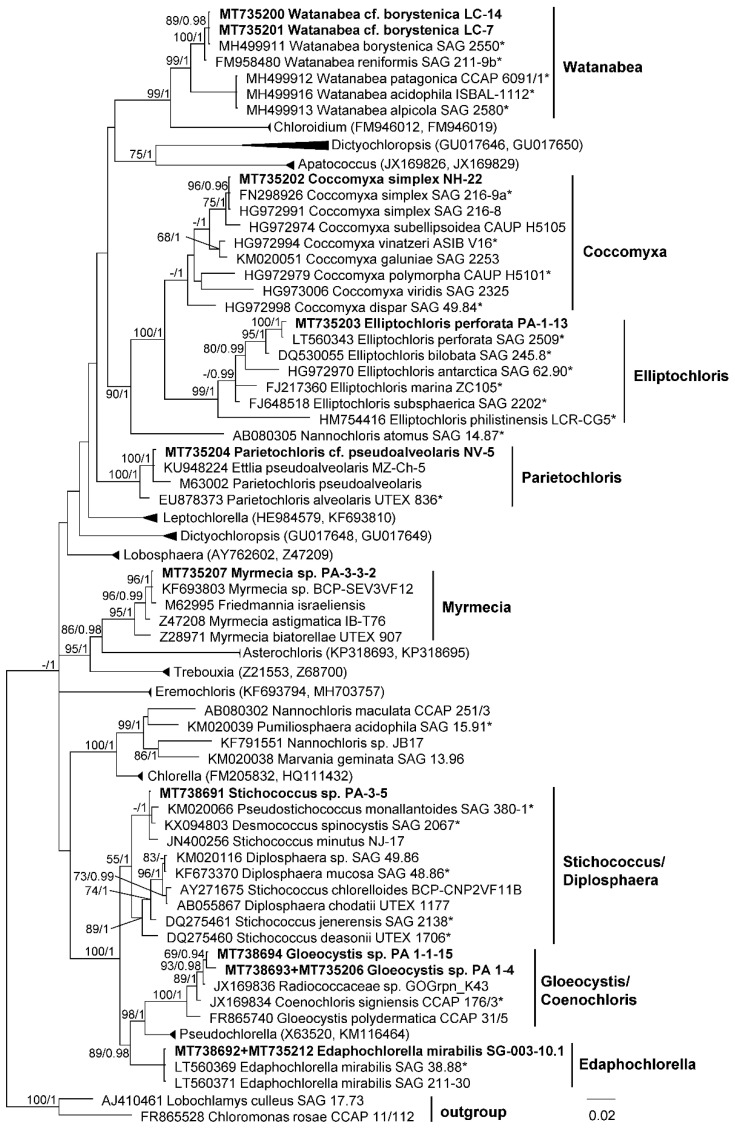
Molecular phylogeny of Trebouxiophyceae based on SSU rRNA sequence analysis. A phylogenetic tree was inferred by the Bayesian method with Bayesian posterior probabilities (PP) and maximum likelihood bootstrap support (BP); branches supported in both analyses (Bayesian values > 0.9 and bootstrap values > 60%) are labeled. Strains marked with an asterisk are authentic strains; strains in bold represent newly sequenced algae.

**Figure 4 microorganisms-08-01047-f004:**
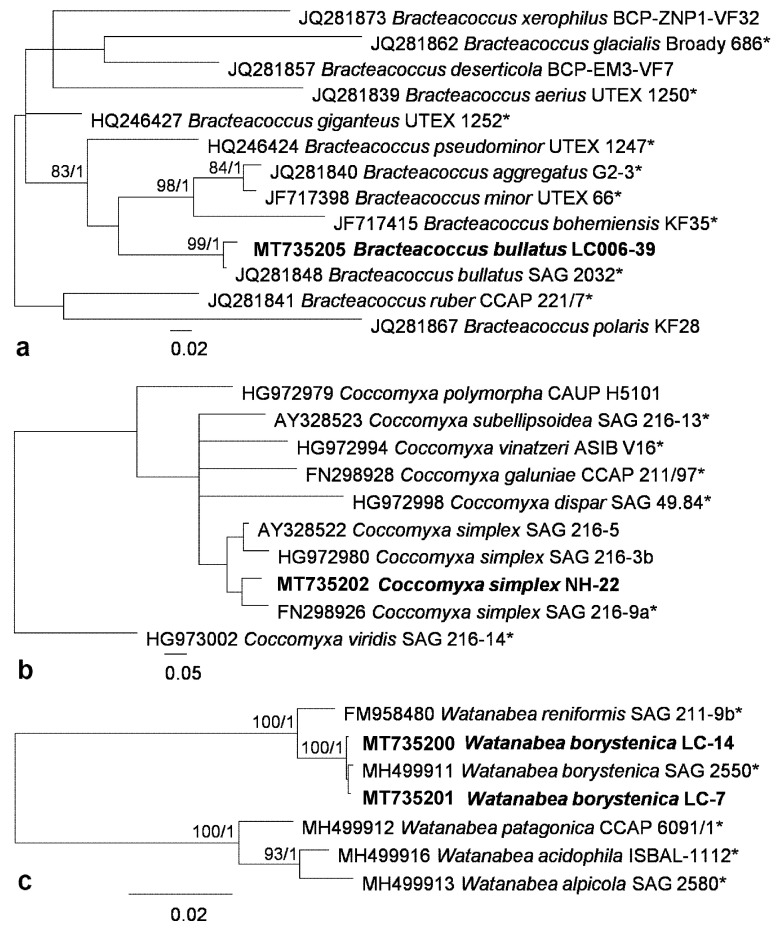
Molecular phylogeny based on ITS region: (**a**) Bracteacoccus (ITS 2), (**b**) Coccomyxa (ITS 2) and (**c**) Watanabea (SSU–ITS 1, 2). Phylogenetic trees were inferred by the Bayesian method with Bayesian posterior probabilities (PP) and maximum likelihood bootstrap support (BP); branches supported in both analyses (Bayesian values > 0.9 and bootstrap values > 60%) are labeled. Strains marked with an asterisk are authentic strains; strains in bold represent newly sequenced algae.

**Figure 5 microorganisms-08-01047-f005:**
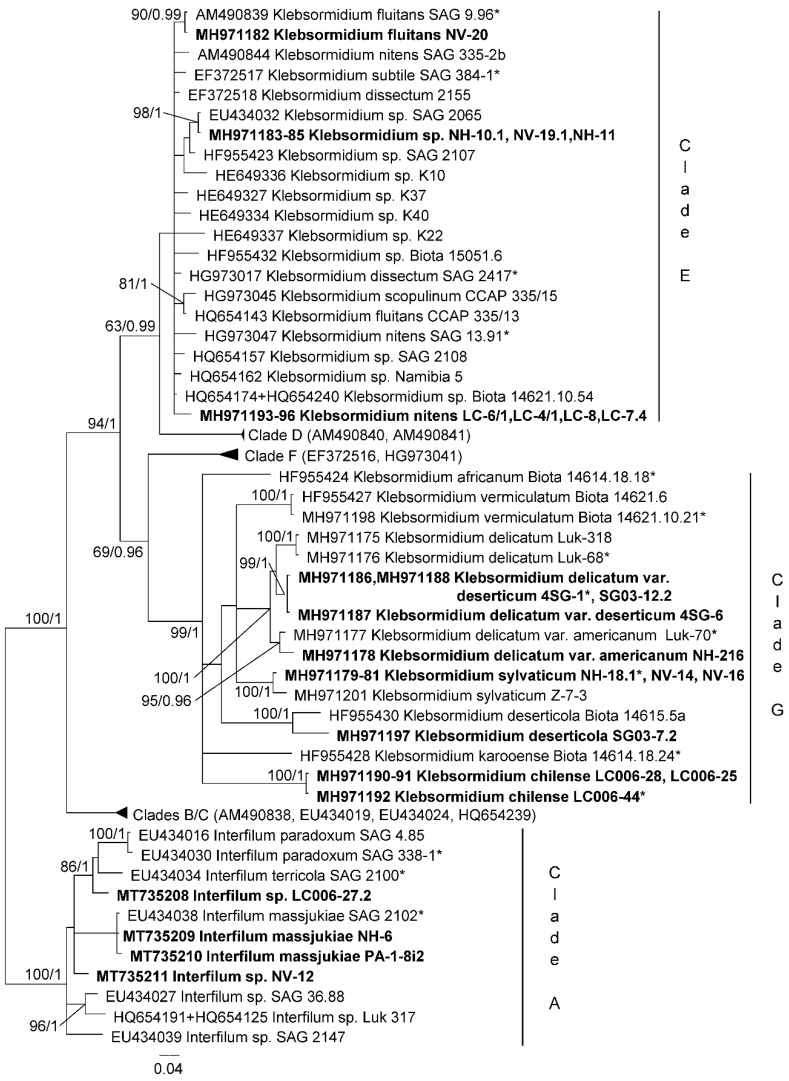
Molecular phylogeny of Klebsormidiophyceae based on ITS-1,2 sequence analysis. A phylogenetic tree was inferred by the Bayesian method with Bayesian posterior probabilities (PP) and maximum likelihood bootstrap support (BP); branches supported in both analyses (Bayesian values > 0.9 and bootstrap values > 60%) are labeled. Strains marked with an asterisk are authentic strains; strains in bold represent newly sequenced algae.

**Figure 6 microorganisms-08-01047-f006:**
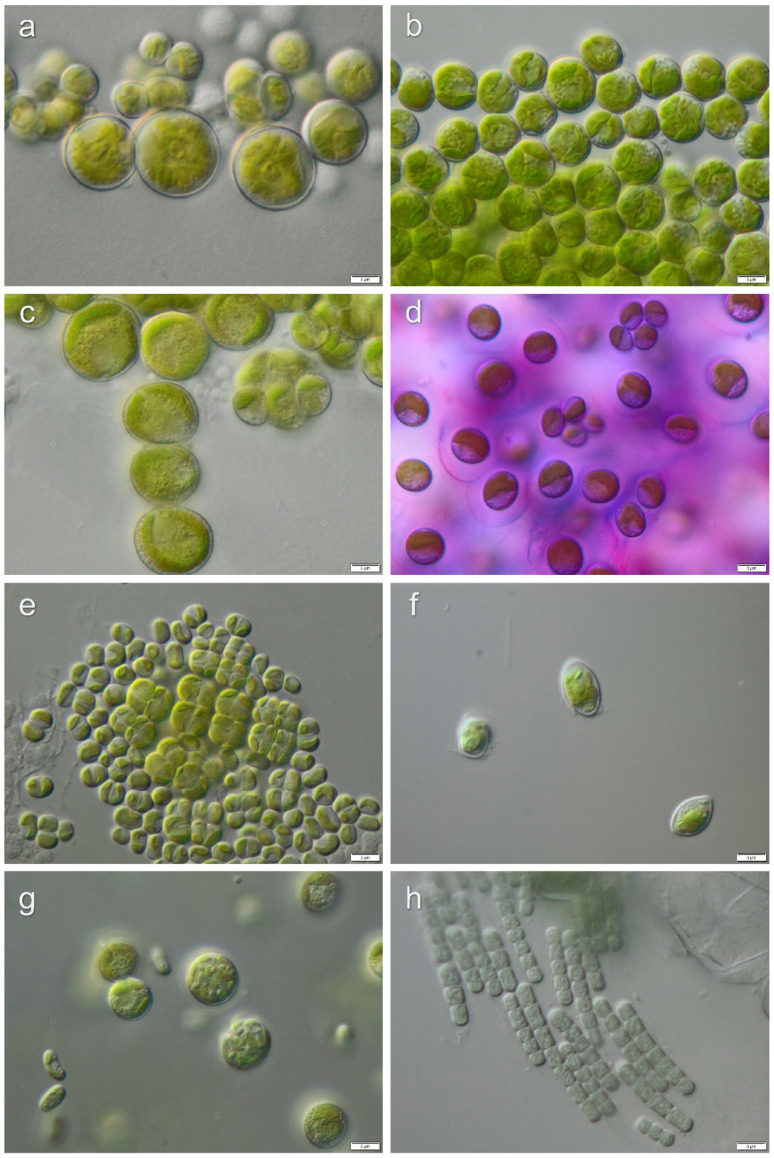
Micrographs of selected algae and cyanobacteria forming Pan de Azúcar biocrusts: (**a**) *Trebouxia* sp., (**b**) *Lobosphaera incisa*, (**c**) *Myrmecia astigmatica*, (**d**) *Gloeocystis* sp., (**e**) *Diplosphaera chodatii*, (**f**) *Chloroidium* sp., (**g**) *Elliptochloris perforata*, (**h**) *Phormidesmis* sp. Scale bars: 5 µm.

**Figure 7 microorganisms-08-01047-f007:**
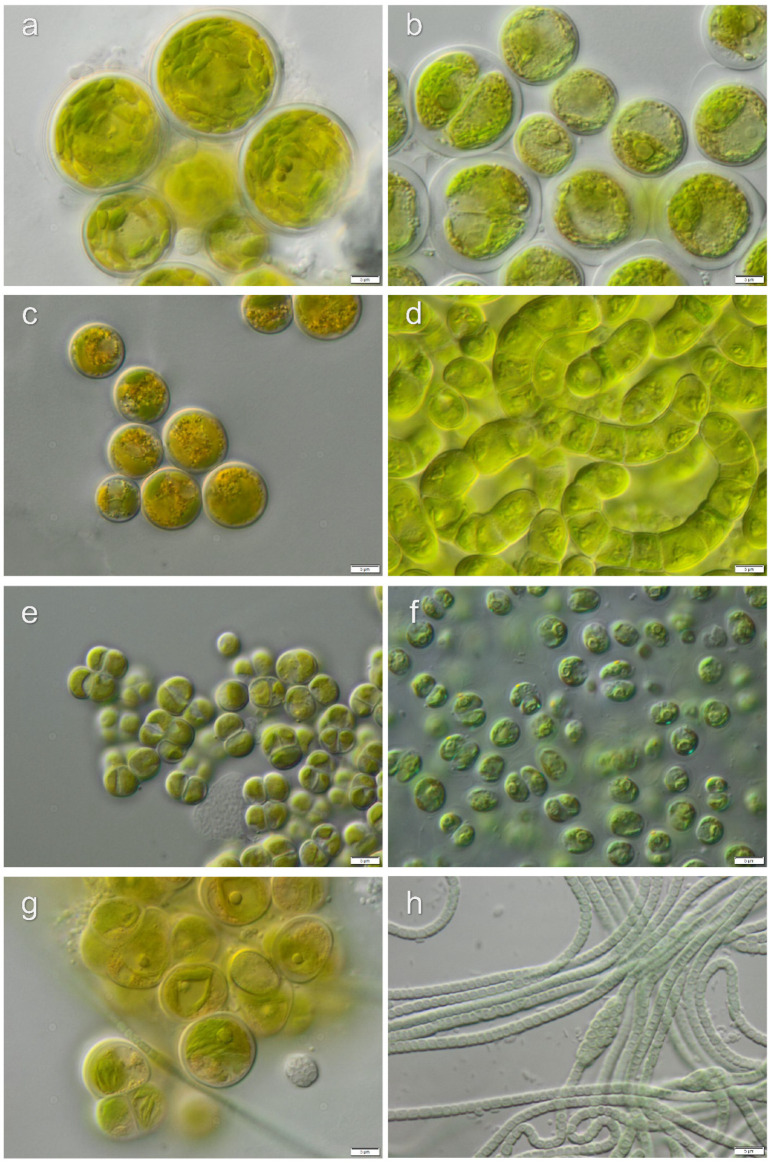
Micrographs of selected algae and cyanobacteria forming Santa Gracia biocrusts: (**a**) *Botrydiopsis* cf. *intercedens*, (**b**) *Chlorococcum* sp.1, (**c**) *Vischeria magna*, (**d**) *Klebsormidium delicatum* var. *deserticum*, (**e**) *Diplosphaera chodatii*, (**f**) *Heterochlamydomonas* sp., (**g**) *Tetracystis* sp., (**h**) *Nodosilinea epilithica.* Scale bars: 5 µm.

**Figure 8 microorganisms-08-01047-f008:**
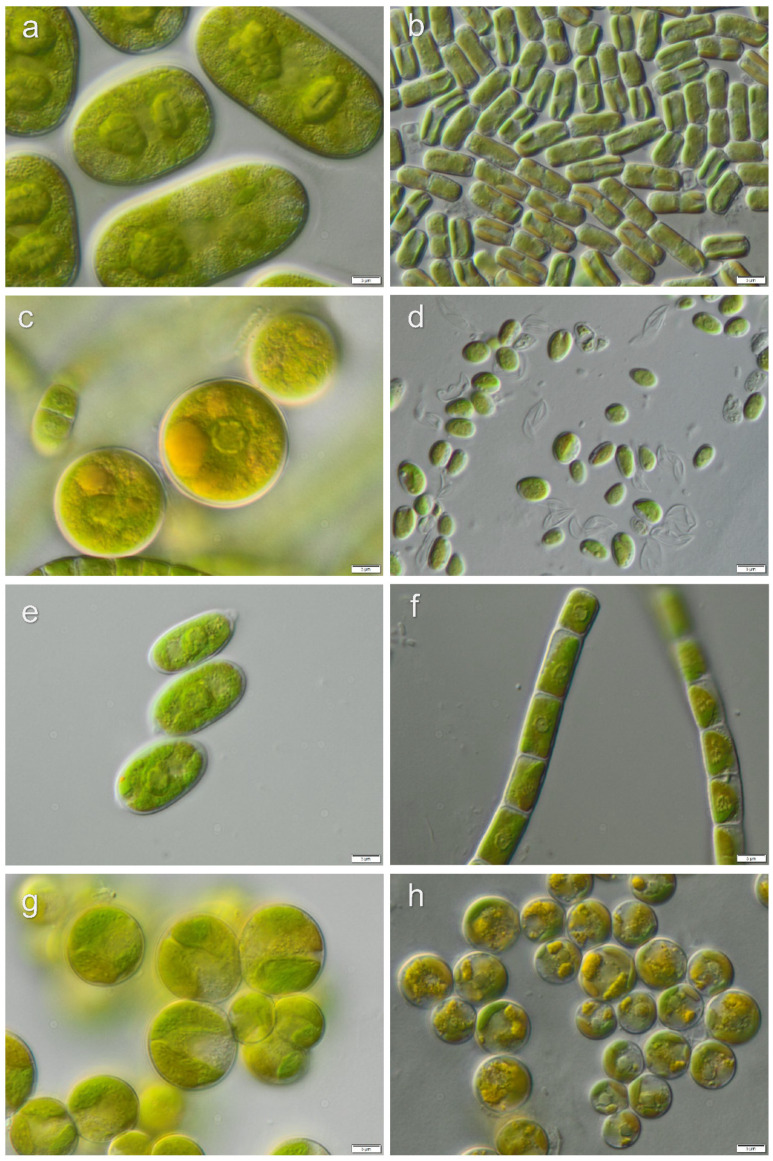
Micrographs of selected algae forming La Campana biocrusts: (**a**) *Cylindrocystis crassa*, (**b**) *Xantonema exile*, (**c**) *Chlorococcum* cf. *oleofaciens*, (**d**) *Xerochlorella minuta*, (**e**) *Lobochlamys culleus*, (**f**) *Klebsormidium nitens*, (**g**) *Myrmecia* cf. *bisecta*, (**h**) *Vischeria vischeri*. Scale bars: 5 µm.

**Figure 9 microorganisms-08-01047-f009:**
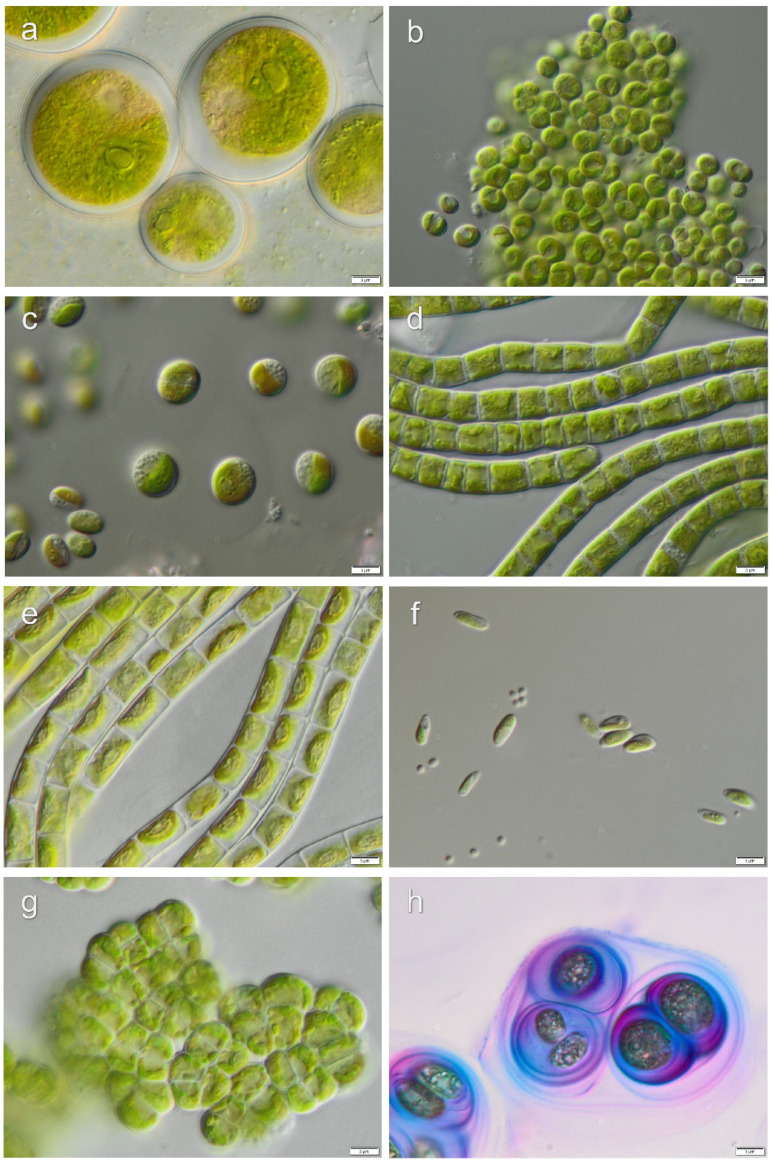
Micrographs of selected algae forming Nahuelbuta biocrusts: (**a**) *Chlorococcum* sp.2, (**b**) *Neocystis* cf. *brevis*, (**c**) *Gloeocystis* cf. *vesiculosa*, (**d**) *Klebsormidium sylvaticum*, (**e**) *Klebsormidium fluitans*, (**f**) *Coccomyxa simplex*, (**g**) *Interfilum* sp., (**h**) *Chlamydomonas* sp.1. Scale bars: 5 µm.

**Figure 10 microorganisms-08-01047-f010:**
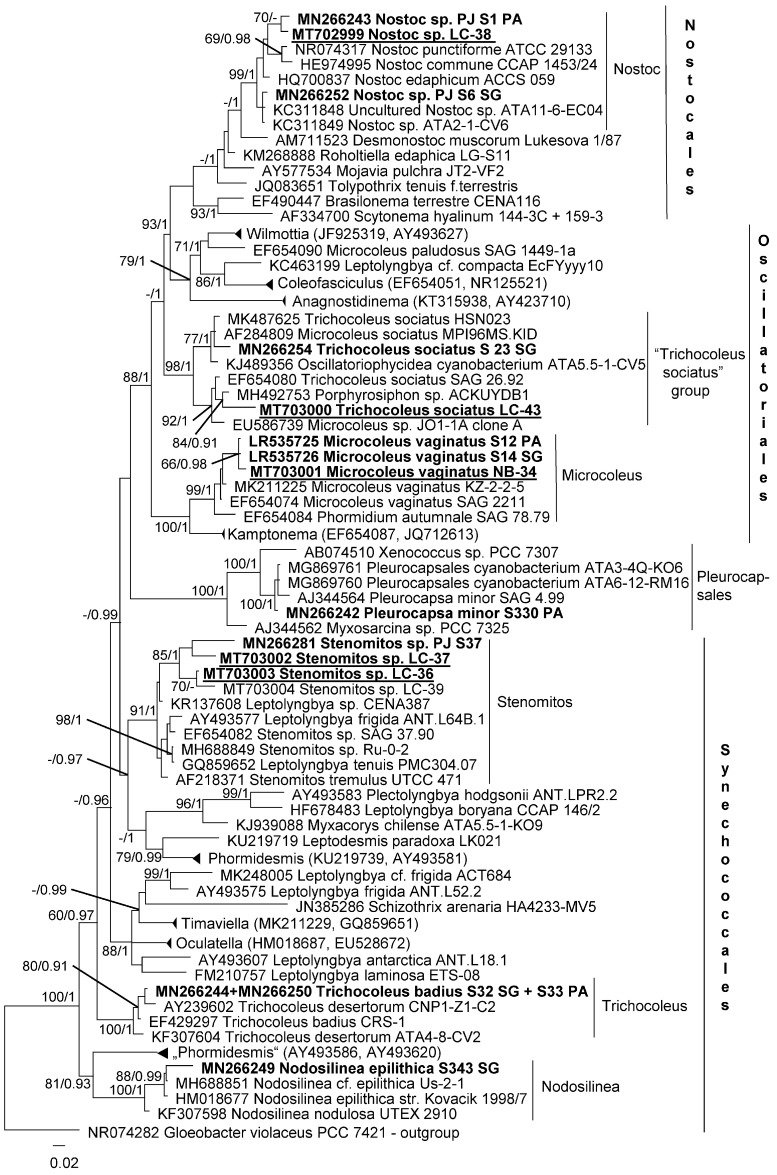
Molecular phylogeny of Cyanobacteria based on SSU rRNA sequence analysis. A phylogenetic tree was inferred by the Bayesian method with Bayesian posterior probabilities (PP) and maximum likelihood bootstrap support (BP); branches supported in both analyses (Bayesian values > 0.9 and bootstrap values > 60%) are labeled. Strains in bold represent newly sequenced cyanobacteria; underlined strains represent Chilean strains together with strains of edaphic cyanobacteria described by Jung et al. [51].

**Figure 11 microorganisms-08-01047-f011:**
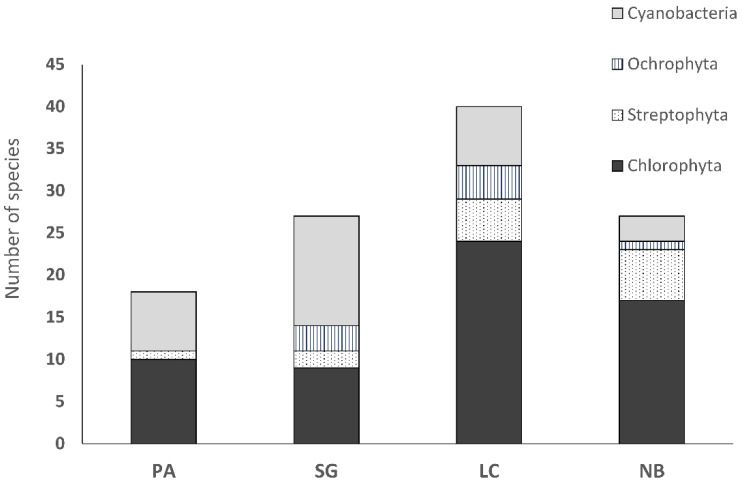
Species richness diagram of biocrusts collected at the 4 sampling locations: PA (Pan de Azúcar), SG (Santa Gracia), LC (La Campana) and NB (Nahuelbuta).

**Table 1 microorganisms-08-01047-t001:** Species composition of eukaryotic algae from biological soil crusts of Chile.

Species	Localities			
	NB	LC	SG	PA
Chlorophyta—45 species				
Chlorophyceae—23 species				
***Bracteacoccus bullatus***	−	**1**	−	−
*Bracteacoccus medionucleatus*	−	1	−	−
***Bracteacoccus* sp.**	**1**	1	−	−
*Chlamydomonas* cf. *pseudoelegans*	−	1	−	−
*Chlamydomonas* sp.1	1	−	−	−
*Chlamydomonas* sp.2	−	1	−	−
*Chlorococcum* cf. *minimum*	−	1	−	−
*Chlorococcum* cf. *minutum*	−	1	−	−
*Chlorococcum* cf. *oleofaciens*	−	1	−	−
*Chlorococcum echinozygotum*	−	1	−	−
*Chlorococcum* sp.1	−	−	1	1
*Chlorococcum* sp.2	1	−	−	−
*Chlorosarcinopsis* cf. *gelatinosa*	−	−	1	−
***Fasciculochloris* sp.**	−	1	**1**	−
***Heterochlamydomonas* cf. *inaequalis***	−	**1**	−	−
***Heterochlamydomonas* sp.**	−	−	**1**	−
***Ixipapillifera* sp.**	−	**1**	−	−
***Lobochlamys segnis***	**1**	**1**	−	−
*Macrochloris* sp.	1	−	−	−
*Neospongiococcum* cf. *excentricum*	1	−	−	−
***Oogamochlamydinia* gen. sp.**	−	**1**	−	−
*Pseudomuriella aurantiaca*	1	−	−	−
***Tetracystis intermedia***	−	−	**1**	−
	7	14	5	1
Trebouxiophyceae—22 species				
*Chlorella* sp.	1	1	−	−
*Chloroidium* sp.	−	−	−	1
***Coccomyxa simplex***	**1**	−	−	−
*Desmococcus* sp.	−	−	1	−
*Diplosphaera chodatii*	1	1	1	1
***Edaphochlorella mirabilis***	−	−	1	−
***Elliptochloris perforata***	−	−	−	**1**
*Elliptochloris subsphaerica*	1	1	1	−
*Gloeocystis* cf. *vesiculosa*	1	1	−	−
***Gloeocystis* sp.**	−	−	−	**1**
*Keratococcus raphidioides*	−	1	−	−
*Leptosira* cf. *erumpens*	−	1	−	−
*Lobosphaera incisa*	1	1	−	1
*Myrmecia* cf. *astigmatica*	−	−	−	1
*Myrmecia* cf. *bisecta*	1	1	−	−
***Myrmecia* sp.**	1	−	−	**1**
*Neocystis* cf. *brevis*	1	−	−	−
***Parietochloris* cf. *pseudoalveolaris***	**1**	−	−	−
***Stichococcus* sp.**	−	−	−	**1**
*Trebouxia* sp.	−	−	−	1
***Watanabea borystenica***	−	**1**	−	−
*Xerochlorella minuta*	−	1	−	−
	10	10	4	9
Streptophyta—12 species				
Klebsormidiophyceae—10 species				
***Interfilum massjukiae***	**1**	−	−	**1**
***Interfilum* sp. 1**	**1**	−	−	−
***Interfilum* sp. 2**	−	**1**	−	−
***Klebsormidium chilense***	−	**1**	−	−
***Klebsormidium delicatum* var. *americanum***	**1**	−	−	−
***Klebsormidium delicatum* var. *deserticum***	−	−	**1**	−
***Klebsormidium deserticola***	−	−	**1**	−
***Klebsormidium fluitans***	**1**	−	−	−
***Klebsormidium nitens***	−	**1**	−	−
***Klebsormidium* sp.**	**1**	−	−	−
***Klebsormidium sylvaticum***	1	−	−	−
	6	3	2	1
Zygnematophyceae—2 species				
*Cylindrocystis brebissonii*	−	1	−	−
*Cylindrocystis crassa*	−	1	−	−
	−	2	−	−
Ochrophyta—5 species				
Xanthophyceae—3 species				
*Botrydiopsis* cf. *constricta*	−	1	−	−
*Botrydiopsis* cf. *intercedens*	−	1	1	−
*Xantonema exile*	−	1	−	−
	−	3	1	−
Eustigmatophyceae—2 species				
*Vischeria magna*	1	−	1	−
*Vischeria vischeri*	−	1	1	−
	1	1	2	−

Species whose original strains were studied by molecular phylogenetic methods are marked in bold. Abreviations: sp.—species, cf.—(lat. conffere) compare with.

**Table 2 microorganisms-08-01047-t002:** Species composition of cyanobacteria from biological soil crusts of Chile.

Species	Localities			
	NB	LC	SG	PA
Chroococcidiopsidales—1 species				
*Chroococcidiopsis* sp.	−	−	−	1
	–	–	–	1
Nostocales—5 species				
*Nostoc* cf. *edaphicum*	−	−	1	−
*Nostoc* cf. *punctiforme*	−	1	1	−
*Nostoc* sp. 1	1	−	−	−
***Nostoc* sp. 2**	−	**1**	−	−
***Nostoc* sp. 3**	−	−	**1**	**1**
	1	2	3	1
Oscillatoriales—8 species				
*Microcoleus* sp.	−	−	1	−
***Microcoleus vaginatus***	**1**	−	**1**	**1**
*Oscillatoria* cf. *tenuis*	−	−	1	−
*Phormidium* sp.	−	−	1	−
*Pseudophormidium* cf. *hollerbachianum*	−	−	−	1
*Myxacorys* sp.1	1	−	−	−
*Myxacorys* sp.2	−	−	1	−
***“Trichocoleus” sociatus***	−	**1**	**1**	−
	2	1	6	2
Pleurocapsales—1 species				
***Pleurocapsa minor***	−	−	−	**1**
	–	–	–	1
Synechococcales—9 species				
*Leptolyngbya henningsii*	−	1	−	−
*Leptolyngbya* sp.	−	−	1	−
*Leptolyngbya tenuis*	−	1	−	−
***Nodosilinea epilithica***	−	−	**1**	−
*Phormidesmis* sp.	−	−	−	1
***Stenomitos* sp. 1**	−	**1**	−	−
***Stenomitos* sp. 2**	−	**1**	−	−
***Stenomitos* sp. 3**	−	−	**1**	−
***Trichocoleus* cf. *badius***	−	−	**1**	**1**
	–	4	4	2

Species whose original strains were studied by molecular phylogenetic methods are marked in Bold.

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
