# Peer review of "Biodiversity of Algae and Cyanobacteria in Biological Soil Crusts Collected Along a Climatic Gradient in Chile Using an Integrative Approach"

_microorganisms, 2020, doi:10.3390/microorganisms8071047_

Round 1

Reviewer 1 Report

In the enclosed file are represented some questions to the authors and some corrections that can be done by authors after which the ms can be accepted.

Author Response

Revision notes microorganisms-823868

We would like to thank the two reviewers for constructive comments on the manuscript. We changed the text and figures according to their suggestions and hope that it now can be accepted for publication in Microorganisms.

Reviewer 1:

In general terms, this is a correctly executed work, on a technical level. However, its weakness is the absence of a stronger starting hypothesis regarding the diversity and/or distribution of the different species found along the studied climatic gradient, which is also not well resolved throughout the discussion. Also, although the authors describe the work as a comprehensive analysis combining molecular and morphological characterization, results section does not describe the morphological characterization, especially needed for those strains of which the authors do not report phylogenetic analysis. On the other hand, several inaccuracies can be detected in the terminology used, which are detailed below.

RE: We provided a stronger starting hypothesis for the study. It now reads (line 115): “We hypothesized that the precipitation gradient shaped the biocrust communities of photosynthetic microorganisms, i.e. the drier the habitat the lower the species richness.”

The most significant is where the authors speak of "diversity of algae and cyanobacteria in biocrusts" based on their results, which are derived from isolates. Although this paper emphasizes the importance of both morphological and molecular characterization of isolates and cultures, it would be a mistake to assume that those cultivable strains represent the whole diversity in situ, since no data (neither microscopy nor molecular) are provided on what is found directly in the biocrusts samples.

RE: We agree with the reviewer that the culture-dependent approach does not represent in-situ diversity in total. Therefore we provided information on limitation of culture-dependent approach in the 4. Discussion line (line 425): “It should be taken into account that in our study community composition and species richness assessment is based on the culture-dependent approach, which can underestimate algal and cyanobacterial diversity in natural communities. The culture-depended approach can potentially lead to the overestimation of algal and cyanobacterial species, which can grow fast in culture. It can also lead to underestimation or even failing to detect uncultivable species, which could be dominant elements in natural biocrusts [19,36].”

Regarding the Introduction, the knowledge gap in biocrusts in the studied area is well supported. However, the authors should consider using more accurate expressions regarding the diversity as a factor increasing stability and productivity of communities (lines 48-56). The manuscript is focused exclusively in the phototrophic members (algae and cyanobacteria) of biocrust communities which share one essential functional trait (primary production). In this context, the introduction does not clarify why a higher diversity in primary producers in biocrusts would be advantageous.

RE: We rewrote the section to express focus of the statement on the phototrophic organisms in a community, it now reads (line 50): “Since phototrophic microorganisms have different, species-specific ecophysiological requirements, i.e. different tolerances against environmental factors, with higher biodiversity the community will have broader tolerance to various abiotic conditions. Also, a higher number of species can simultaneously provide multiple functions in a biocrust, which will result in the more productive community [6,7]”

Again, in regarding to the accuracy of the used terms, the authors cited the work of Lhotsky (1998) in line 57, talking about “microalgal species”. It seemed that the authors used that citation as data concerning only to algae, while in Lhotský (1998) the author used "microalgal species" referring to both algae and cyanobacteria, which is not correct currently.

RE: We excluded the statement from the manuscript since the statement refers to aquatic organisms.

The same kind of inaccuracy can be observed in lines 62-66, where the authors explain how “water-holding mechanisms (…) guarantee more efficient dispersal compared to their aquatic counterparts”, which is not necessarily true, since these mechanisms facilitate not dispersion, as such, but more likely the ability to colonize other environments where water is less available.  

RE: In the statement, we were referring to the ability of aero-terrestrial algae and cyanobacteria to survive unfavourable conditions while being transported in aero-terrestrial conditions, as stated in Sharma et al. (2007): “Airborne algae experience severe dehydration stress, especially immediately after being aerosolized (Ehresmann and Hatch 1975). This phenomenon adversely affects their survival. It is likely that the algae adapted to terrestrial habitats are more resistant to a state of dehydration and, thus, better adapted to aerial dispersal than their aquatic counterparts (Schlichting 1969, Ehresmann and Hatch 1975)”. Thus the organisms’ mechanisms to ensure desiccation protection directly contribute to dispersion and colonisation of the new environment.

In order to clarify that the statement refers to the aero-terrestrial algae and cyanobacteria in the aero-terrestrial environment, we rewrote the statement. It now reads (line 58): “Microalgae and cyanobacteria can be transported to varying distances by air or by moving animals (birds, insects, etc.), yet an efficient dispersal in aero-terrestrial conditions also depends on the resistance of species to survive unfavourable conditions during transport (e.g. desiccation and intensive UV radiation [10]).”

The last paragraph of the introduction (lines 112-117) properly expresses the importance and interest of this work, as the first study that presents an extensive characterization of the primary producers of biocrusts in Chile. However, in spite of the interesting title of the work that refers to the climatic gradient, the presentation of a starting hypothesis in this respect is missing, seeming to present a purely descriptive work. For this reason, the presence of a sentence that expresses what the authors expected from the laborious analysis they have carried out would be appreciated.

RE: We stated the objective of the study. It now reads (line 112): “Since all previous studies of terrestrial cyanobacteria and eukaryotic algae were focused exclusively on the arid area of the Atacama Desert, the objective of present study was to document in detail the diversity of key biocrust photosynthetic organisms in study areas which follow the precipitation gradient along the Pacific coast of Chile.”

Such as when the authors took the samples and how were they kept until the analysis since those conditions could improve the physiological status of some members in relation to others, posing a problem for their future chances of being cultivated.

RE: We added the information on the sampling time, as well as the details how the samples were stored until the analysis. It now reads:

line 159: “Biocrust material was collected during sampling campaigns in January and March 2016.”

line 164: “The collected material was air-dried for the transport, then kept in the laboratory at room temperature in the dark. Biocrust material was used for the establishment of enrichment cultures within three weeks after the sampling.”

To know the date when NCBI was consulted in order to compare the sequences (line 204) would be also interesting.

RE: We provided the date when NCBI was consulted in order to compare the sequences (line 213): “the NCBI database was searched for algae on 22. January and for cyanobacteria on 17. February 2020.”

Also, the quality of figure 1 is too poor to be correctly read.

RE: The Figure 1. is presented in better quality.

Section 3. Results is clearly exposed and the high-quality of microscopy images is appreciated. The missing of phylogenetic characterization of some of the isolates is exhibited although no explanation for it can be found in the manuscript. The results concerning the morphological characterization are missing as well, such as features that have been taken into consideration and images that support the taxonomic assignment given by the authors, especially for those isolates without molecular characterization, previously mentioned.

RE: We provided information on the identification methodology, together with the previously missing information on scientific literature used for the identification of cyanobacteria. It now reads (line187): “Morphological identification of eukaryotic algae was based on Ettl & Gärtner [39] and the latest scientific literature on specific taxa; morphological identification of cyanobacteria was based on Komárek & Anagnostidis [40] and Komárek [41] as well as latest scientific literature on specific cyanobacterial taxa.”

Also, some discrepancies can be found in Figure 10 (line 340) between the text and the image, since the text refers to “strains marked with an asterisk are authentic strains” (line 344) and no asterisks can be found in the figure, whereas the figure contains red underlined strains with no explanation in the text.

Re: The discrepancies between Figure 10. and its caption was corrected. The caption now reads (line 352): “Molecular phylogeny of Cyanobacteria based on SSU rRNA sequence analysis. A phylogenetic tree was inferred by the Bayesian method with Bayesian Posterior Probabilities (PP) and Maximum Likelihood bootstrap support (BP); branches supported in both analyses (Bayesian values > 0.9 and bootstrap values > 60%) are labelled. Strains in bold represent newly sequenced cyanobacteria; underlined strains represent Chilean strains together with strains of edaphic cyanobacteria described by Jung et al. [51].”

A problem of a conceptual character has been found in section 3.2. Biocrust Community Composition and Species Richness (line 367 and following), since, as explained at the beginning of this review, these are data concerning isolates. For instance, the authors state that “Desert communities from Pan de Azúcar (PA) were composed exclusively of..." (line 368) which is highly inaccurate, since the information available to them is only on those species that they have managed to isolate and cultivate. The greater or lesser richness and/or diversity of the different samples could have been discussed as representative or not of what is in the field, but it is not correct to say that what was isolated represented fully the reality of the biocrusts in the field.  Throughout the whole section, the authors talk about the diversity of the biocrusts, which is incorrect based on the data they are working with. Thus, rewriting of this section, again, taking into account the limitations of the obtained results would be appreciated.

RE: We specified that we applied the culture-dependent approach for the identification and species richness assessment of algae and cyanobacteria in section 4. Discussion. It now reads (line 381): “With regard to the culture-based community composition and species richness assessment, desert biocrusts from Pan de Azúcar (PA) were composed… ”.

The application of the culture-dependent approach was stated in section 3. Results (sections 3.1. and 3.2.), line 236 and 344.

Regarding the section 4. Discussion it is difficult to understand. The main weakness, as it already happened in the results section, is that it seems that the authors assume the characterization of isolates as the characterization of the entire phototrophic fraction of the biocrusts communities. This issue should be rewrite and reconsider taking into account only the limitation of working with culture-depending data. Also, a deeper discussion on the effect of the precipitation gradient on the presence/absence of different taxa in the biocrusts would be highly appreciated (sections 4.1 and 4.2).

RE: We rewrote the section, it now reads (line 426): “It should be taken into account that in our study community composition and species richness assessment is based on the culture-dependent approach, which can underestimate algal and cyanobacterial diversity in natural communities. The culture-depended approach can potentially lead to the overestimation of algal and cyanobacterial species, which can grow fast in culture. It can also lead to underestimation or even failing to detect uncultivable species, which could be dominant elements in natural biocrusts [19, 36].”

Some minor comments on the discussion: The authors should be careful again with the use of inaccurate or too colloquial vocabulary (line 506 – “Chroococcidiopsis sp. prefers” as an example). The authors forget to mention Chroococcidiopsis sp. as the main phototrophic member also in endolithic communities in Chile (Dong et al. 2007, Cámara et al. 2015, Wierzchos et al. 2018, as examples) (line 508).

RE: We replaced inaccurate and too colloquial vocabulary with appropriate expressions, we also included the information on the presence of Chroococcidiopsis sp. as the main phototrophic member in endolithic communities in Chile, followed by the suggested references. It now reads (line 524): “Cyanobacteria found in PA biocrusts include coccoid Chroococcidiopsis sp., which was reported as a constituent of biocrust communities worldwide [55] as well as a member of hypolithic communities in Chile [71–73]”.

Also, sometimes it is difficult to understand where the authors are referring to their own results or to previous studies (lines 419-426, as an example).

RE: We rewrote the sentence; it now reads (line 432): “Biocrust communities we analysed were predominately composed of algae from the classes Chlorophyceae and Trebouxiophyceae, followed by Klebsormidiophyceae, which corresponds to previous studies on biocrusts from Europe, Asia, Africa, North America, Australia, and Polar Regions of Russia presented by Büdel et al. [19].”

Conclusions section, on the contrary, summarized perfectly the real and interesting impact of the data the authors are providing in this work, not referring to the inaccuracies and confusing ideas occurring in the discussion section. 

RE: We included the information of limitations of the culture-dependent approach to represent the diversity of organisms in natural communities. It now reads (line 546): “It should be taken into account that our study is based on the culture-dependent approach, which can underestimate the algal and cyanobacterial diversity present in natural communities.”

Reviewer 2 Report

The manuscript entitled as “Biodiversity of Algae and Cyanobacteria in Biological Soil Crusts Collected Along a Climatic Gradient in Chile Using an Integrative Approach” tries to show the diversity of algae and cyanobacteria from biocrusts in such an understudied region as Chile (South America). Also, it tries to find how different climatic conditions determine different species composition of these biocrusts. This work combines different techniques to study both algae and cyanobacteria from those biocrusts.

The strength of this work is undoubtedly the large number of strains isolated from the different biocrusts, giving light to the knowledge gap that the authors describe well in the introduction. Also remarkable, is the high quality of the microscopy images provided, which is highly appreciated.

In general terms, this is a correctly executed work, on a technical level. However, its weakness is the absence of a stronger starting hypothesis regarding the diversity and/or distribution of the different species found along the studied climatic gradient, which is also not well resolved throughout the discussion. Also, although the authors describe the work as a comprehensive analysis combining molecular and morphological characterization, results section does not describe the morphological characterization, especially needed for those strains of which the authors do not report phylogenetic analysis. On the other hand, several inaccuracies can be detected in the terminology used, which are detailed below. The most significant is where the authors speak of "diversity of algae and cyanobacteria in biocrusts" based on their results, which are derived from isolates. Although this paper emphasizes the importance of both morphological and molecular characterization of isolates and cultures, it would be a mistake to assume that those cultivable strains represent the whole diversity in situ, since no data (neither microscopy nor molecular) are provided on what is found directly in the biocrusts samples.

Regarding the Introduction, the knowledge gap in biocrusts in the studied area is well supported. However, the authors should consider using more accurate expressions regarding the diversity as a factor increasing stability and productivity of communities (lines 48-56). The manuscript is focused exclusively in the phototrophic members (algae and cyanobacteria) of biocrust communities which share one essential functional trait (primary production). In this context, the introduction does not clarify why a higher diversity in primary producers in biocrusts would be advantageous. Again, regarding to the accuracy of the used terms, the authors cited the work of Lhotsky (1998) in line 57, talking about “microalgal species”. It seemed that the authors used that citation as data concerning only to algae, while in Lhotský (1998) the author used "microalgal species" referring to both algae and cyanobacteria, which is not correct currently. The same kind of inaccuracy can be observed in lines 62-66, where the authors explain how “water-holding mechanisms (…) guarantee more efficient dispersal compared to their aquatic counterparts”, which is not necessarily true, since these mechanisms facilitate not dispersion, as such, but more likely the ability to colonize other environments where water is less available.   

The last paragraph of the introduction (lines 112-117) properly expresses the importance and interest of this work, as the first study that presents an extensive characterization of the primary producers of biocrusts in Chile. However, in spite of the interesting title of the work that refers to the climatic gradient, the presentation of a starting hypothesis in this respect is missing, seeming to present a purely descriptive work. For this reason, the presence of a sentence that expresses what the authors expected from the laborious analysis they have carried out would be appreciated.

Although section 2. Material and Methods is properly described some details are missing. Such as when the authors took the samples and how were they kept until the analysis, since those conditions could improve the physiological status of some members in relation to others, posing a problem for their future chances of being cultivated. To know the date when NCBI was consulted in order to compare the sequences (line 204) would be also interesting. Also, the quality of figure 1 is too poor to be correctly read.

Section 3. Results is clearly exposed and the high-quality of microscopy images is appreciated. The missing of phylogenetic characterization of some of the isolates is exhibited although no explanation for it can be found in the manuscript. The results concerning the morphological characterization are missing as well, such as features that have been taken into consideration and images that support the taxonomic assignment given by the authors, especially for those isolates without molecular characterization, previously mentioned.

Also, some discrepancies can be found in Figure 10 (line 340) between the text and the image, since the text refers to “strains marked with an asterisk are authentic strains” (line 344) and no asterisks can be found in the figure, whereas the figure contains red underlined strains with no explanation in the text. 

A problem of a conceptual character has been found in section 3.2. Biocrust Community Composition and Species Richness (line 367 and following), since, as explained at the beginning of this review, these are data concerning isolates. For instance, the authors state that “Desert communities from Pan de Azúcar (PA) were composed exclusively of..." (line 368) which is highly inaccurate, since the information available to them is only on those species that they have managed to isolate and cultivate. The greater or lesser richness and/or diversity of the different samples could have been discussed as representative or not of what is in the field, but it is not correct to say that what was isolated represented fully the reality of the biocrusts in the field.  Throughout the whole section, the authors talk about the diversity of the biocrusts, which is incorrect based on the data they are working with. Thus, rewriting of this section, again, taking into account the limitations of the obtained results would be appreciated.

Regarding the section 4. Discussion it is difficult to understand. The main weakness, as it already happened in the results section, is that it seems that the authors assume the characterization of isolates as the characterization of the entire phototrophic fraction of the biocrusts communities. This issue should be rewrite and reconsider taking into account only the limitation of working with culture-depending data. Also, a deeper discussion on the effect of the precipitation gradient on the presence/absence of different taxa in the biocrusts would be highly appreciated (sections 4.1 and 4.2).

Some minor comments on the discussion: The authors should be careful again with the use of inaccurate or too colloquial vocabulary (line 506 – “Chroococcidiopsis sp. prefers” as an example). The authors forget to mention Chroococcidiopsis sp. as the main phototrophic member also in endolithic communities in Chile (Dong et al. 2007, Cámara et al. 2015, Wierzchos et al. 2018, as examples) (line 508). Also, sometimes it is difficult to understand where the authors are referring to their own results or to previous studies (lines 419-426, as an example).

Conclusions section, on the contrary, summarized perfectly the real and interesting impact of the data the authors are providing in this work, not referring to the inaccuracies and confusing ideas occurring in the discussion section.  

Author Response

Revision notes microorganisms-823868

We would like to thank the reviewer for constructive comments on the manuscript. We changed the text and figures according to their suggestions and hope that it now can be accepted for publication in Microorganisms.

Reviewer 2:

1. We referred to Streptophyta as currently used synonym for Charophyta.

(see: Adl, Sina M., David Bass, Christopher E. Lane, Julius Lukeš, Conrad L. Schoch, Alexey Smirnov, Sabine Agatha et al. "Revisions to the classification, nomenclature, and diversity of eukaryotes." Journal of Eukaryotic Microbiology 66, no. 1 (2019): 4-119.)

2. We specified that the “Mediterranean” refers to climate zone, we also provided altitude for every sampling location. The sentence now reads (line 123): “… with National Park Pan de Azúcar (PA) in the arid north at 651 m above sea level (a.s.l.) , Nature Reserve Santa Gracia (SG) in the semi-arid zone at 706 m a.s.l., National Park La Campana (LC) in the Mediterranean climate zone at 726 m a.s.l., and National Park Nahuelbuta (NB) at 770 m a.s.l. in the temperate-humid zone of Chile”

3. We corrected Figure 1. caption, it now reads (line128): “Shown are the monthly average values of diurnal temperature (°C) represented with blue bars and precipitation (mm) represented with red line; months are represented numerically. Source: climatecharts.net.”

4. We corrected Figure 8. caption, it now reads (line 326): “Micrographs of selected algae forming La Campana biocrusts: (A) Cylindrocystis crassa, (B) Xantonema exile, (C) Chlorococcum oleofaciens, (D) Xerochlorella minuta, (E) Lobochlamys culleus, (F) Klebsormidium nitens, (G) Myrmecia cf. bisecta, (H) Vischeria vischeri. Scale bars: 5 µm”

5. We corrected Figure 9. caption, it now reads (332): “(H) Lobochlamys sp.

6. We differentiated filamentous algae from filamentous cyanobacteria. It now reads:

line 491: “Therefore, filamentous algae and filamentous cyanobacteria were present in biocrusts in all other regions.” 

line 493: “In semi-arid SG, filamentous cyanobacteria were prevailing over algae, contrary to Mediterranean LC biocrusts, which were predominantly composed of eukaryotic algae, followed by filamentous and colonial cyanobacteria.”

line 497: “While filamentous cyanobacteria dominated biocrusts in the semi-arid region, eukaryotic filamentous algae had the highest richness in the temperate region.”

7. We corrected Unfavourable corrected to unfavorable (line 499): “…can be attributed to unfavorable soil pH and…” .